# Indocyanine Green Angiography for Parathyroid Gland Evaluation during Transoral Endoscopic Thyroidectomy

**DOI:** 10.3390/jpm11090843

**Published:** 2021-08-27

**Authors:** Tsung-Jung Liang, Kuo-Chiang Wang, Nai-Yu Wang, I-Shu Chen, Shiuh-Inn Liu

**Affiliations:** 1Division of General Surgery, Department of Surgery, Kaohsiung Veterans General Hospital, Kaohsiung 81362, Taiwan; tjliangmd@gmail.com (T.-J.L.); wangkc@vghks.gov.tw (K.-C.W.); greenmilk0627@gmail.com (N.-Y.W.); nugaticc@gmail.com (I.-S.C.); 2School of Medicine, National Yang Ming Chiao Tung University, Taipei 11221, Taiwan

**Keywords:** indocyanine green, fluorescence, transoral endoscopic thyroidectomy, parathyroid, thyroid

## Abstract

Indocyanine green (ICG) angiography, a real-time intraoperative imaging technique, is associated with better parathyroid identification and functional evaluation during open thyroidectomy. However, the benefits of ICG fluorescence imaging application in transoral endoscopic thyroidectomy are not well-documented. Consecutive patients who underwent transoral endoscopic thyroidectomy were retrospectively reviewed. Parathyroid glands were assessed with visual inspection followed by ICG angiography. The fluorescence intensity of all parathyroid glands was recorded. In total, 158 parathyroid glands from 60 patients (41 underwent lobectomy and 19 underwent total thyroidectomy) were eligible for evaluation. A total of 135 parathyroid glands (85.4%) were identified, including nine glands (5.7%) that were solely localized because of ICG angiography. Incidental parathyroidectomy occurred in 12 patients with predominant inferior gland (83.3%) and associated with central neck dissection (66.7%). Among patients receiving total thyroidectomy, patients who retained at least one well-perfused parathyroid gland had higher parathyroid hormone (PTH) level and were less likely to develop hypoparathyroidism on postoperative day one than those without any well-perfused ICG-enhanced parathyroid gland (*p* = 0.038). In addition, the duration of calcium supplementation to maintain normocalcemia was also shorter. ICG angiography is a feasible adjunct procedure for parathyroid identification and postoperative functional prediction in transoral endoscopic thyroidectomy.

## 1. Introduction

Indocyanine green (ICG) angiography has gained popularity in the field of thyroid and parathyroid surgery recently [1,2]. By capturing the emitting near infrared fluorescence, it offers two kinds of benefits. The first benefit is the identification and localization of parathyroid, so that surgeons can preserve the healthy parathyroid gland, as in the case of thyroidectomy, and remove the diseased parathyroid gland, as in patient with primary hyperparathyroidism [3]. The second benefit is the assessment of the viability of the parathyroid gland, which may predict the postoperative parathyroid function and help in appropriate patient management [4].

In recent years, various kinds of endoscopic and robotic approaches for thyroidectomy and parathyroidectomy have been proposed and have shown equivalent safety with superior cosmetic outcome when compared with the open surgery [5,6,7,8]. Among these techniques, the transoral approach is one of the most popular and promising routes because it requires less flap dissection and leaves no scar on the body surface [9].

Currently, few studies are available that have evaluated the implementation of ICG angiography in endoscopic or robotic thyroidectomy [10,11,12]. Yu et al. analyzed the patients who underwent robotic thyroidectomy via the bilateral axillo-breast approach and found that the patients who had received ICG for parathyroid localization using the Firefly technology had a significantly lower rate of incidental parathyroidectomy [10]. However, their study only evaluated the inferior glands without concurrently assessing the superior glands [10]. With respect to the endoscopic transoral approach, Turan et al. reported the first clinical study applying ICG angiography to localize the parathyroid adenoma in seven patients with hyperparathyroidism [11]. The target parathyroid glands were identified and removed. The parathyroid hormone (PTH) level fell back into normal range in all patients. Nevertheless, their study did not enroll patients with normal parathyroid function and their sample size was relatively small [11].

To assess the benefits of ICG angiography in transoral endoscopic thyroidectomy more thoroughly, we aimed to evaluate the fluorescence intensity of both superior and inferior parathyroid glands. The associated clinical data, including PTH, serum calcium level, and duration of calcium supplementation, were also recorded. Collectively, these objectives were aimed to help us to be able to evaluate the contribution of ICG angiography in parathyroid localization and postoperative functional prediction more clearly.

## 2. Materials and Methods

### 2.1. Patients

Between January 2020 and March 2021, the medical records of all patients who underwent transoral endoscopic thyroidectomy via vestibular approach at Kaohsiung Veterans General Hospital were reviewed. The same surgeon (T.J.L.) performed all the operations. At our institute, ICG angiography was routinely implemented during endoscopic thyroidectomy, unless there was an unavailability of the laparoscopic infrared imaging system. The surgical indications included benign thyroid nodules with diameter <5 cm and well-differentiated thyroid carcinoma without preoperative evidence of lymph node metastasis. The institutional review board of the author’s affiliation approved this study (KSVGH21-CT6-03).

Transoral endoscopic thyroidectomy was performed as described by Anuwong et al. with a modification in the step of working space creation [13]. We applied the Foley balloon to dilate the subplatysmal space so as to minimize the use of blunt dissection and its associated injury [14]. The details of the perioperative management have been delineated in our previous publication [15].

### 2.2. Parathyroid Assessment

We did not intentionally search for parathyroid glands during thyroid dissection. After thyroidectomy, parathyroid glands were identified through visual inspection followed by ICG angiography. For patients with clinical suspicion of malignancy, central neck dissection was performed after parathyroid assessment to avoid inadvertent parathyroid removal along with the lymph nodes.

Visual assessment of the parathyroid gland was performed using a two-point grading system, as proposed by Rudin et al.: score 2 (viable) and score 0 (non-viable) [16]. Then, a single dose of 7.5 mg ICG was administered through peripheral vein and ICG fluorescence was evaluated approximately one minute later by a 30 degree 10 mm infrared (IR) telescope with VISERA ELITE II system (Olympus, Tokyo, Japan) [11]. This imaging system provided two types of IR images: combined partial white light with IR light (IR Mode 1), and pure fluorescent image (IR Mode 2). The fluorescence intensity was graded as 0, 1, and 2, representing the none, partial, and well-vascularized status, respectively [16,17]. Parathyroid glands that lacked vascularity, both by visual and ICG assessment (score 0), were auto-transplanted in the sternocleidomastoid muscle.

### 2.3. Outcome Evaluation and Follow-Up

Incidental parathyroidectomy was defined as the presence of parathyroid tissue in the permanent section. In patients who underwent total thyroidectomy, serum calcium and PTH were routinely checked on postoperative day one, and then one week and one month later. Biochemical hypoparathyroidism was defined as PTH level <15 pg/mL (normal range, 15–68 pg/mL). Calcium and vitamin D were prescribed only to those who developed symptomatic hypocalcemia (normal range, 8.4–10.6 mg/dL). The duration of calcium supplementation that was required to maintain normocalcemia was recorded.

### 2.4. Statistical Analysis

For patients who underwent total thyroidectomy, postoperative outcomes were compared between those who had at least one well-vascularized parathyroid gland (ICG 2 ≥ 1) and those who had no well-vascularized parathyroid gland (ICG 2 = 0). Continuous variables were expressed as mean ± standard deviation and analyzed by Mann-Whitney U test. Categorical variables were compared by Fisher’s exact test. Statistical significance was set at *p* < 0.05. The statistical analyses were performed by SPSS, version 20 (IBM Co., Armonk, NY, USA).

## 3. Results

### 3.1. Demographics

During the study period, 69 consecutive patients were reviewed, and nine patients were excluded due to having no access to the infrared imaging system at the time of their surgery. In total, 60 patients were enrolled (Table 1). Among those, 41 patients (68.3%) underwent lobectomy and 19 patients (31.7%) received total thyroidectomy. As a result, 158 parathyroid glands were eligible for assessment. Twenty-two patients (36.7%) had central neck dissection due to clinical suspicion for malignancy.

### 3.2. Parathyroid Gland Identification

Through initial visual inspection, only 79.7% (126/158) parathyroid glands were identified (Table 2). After implementing the ICG angiography, an additional nine parathyroid glands (one superior and eight inferior) were localized via the fluorescence image (Figure 1), which raised the parathyroid gland detection rate up to 85.4% (135/158).

On pathological examination, 12 parathyroid glands (7.6%) were found in the surgical specimens (Table 3). After correlating with intraoperative assessment results, most of the inadvertently removed parathyroid glands were the inferior ones (10/12, 83.3%) and associated with central neck dissection (8/12, 66.7%). The major cause of incidental parathyroidectomy was the non-identification of the parathyroid glands with both visual inspection and ICG angiography (9/12, 75%). In the remaining three patients (25%), fluorescence of other tissues, such as fat or lymph node, interfered with the detection of parathyroid gland and caused misidentification, which led to incidental removal of the real parathyroid glands.

### 3.3. Parathyroid Glands Perfusion

Table 2 summarizes the assessment results of parathyroid glands perfusion using visual inspection and ICG angiography.

In 19 patients who underwent total thyroidectomy, 14 (73.7%) had at least one well-perfused parathyroid glands (ICG 2 ≥ 1) (one gland, *n* = 6; two glands, *n* = 5; three glands, *n* = 3). In the remaining five patients, all their glands evaluated were either partially vascularized or devascularized without being well-perfused (ICG 2 = 0).

Patients who retained at least one well-perfused parathyroid gland (ICG 2 ≥ 1) had significantly higher PTH levels (Figure 2A), and were less likely to develop hypoparathyroidism on postoperative day one than those without any well ICG-enhanced parathyroid gland (ICG 2 = 0) (*p* = 0.038) (Table 4). Their serum calcium levels on postoperative day one was also higher (Figure 2B). In addition, shorter duration of calcium supplementation was required to maintain normocalcemia in patients with at least one well-perfused parathyroid gland (ICG 2 ≥ 1) (8.0 vs. 18.2 days, *p* = 0.034).

## 4. Discussion

We applied ICG angiography to evaluate parathyroid glands in patients who underwent transoral endoscopic thyroidectomy. Under fluorescence guidance, an additional nine parathyroid glands that were not seen on visual inspection were identified, which might lower the risk of incidental parathyroidectomy during thyroid surgery. Patients with at least one well-perfused, fluorescence-enhanced parathyroid gland (ICG 2 ≥ 1) had significantly higher postoperative PTH levels, lower chances of hypoparathyroidism, and required shorter durations of calcium supplementation to maintain normocalcemia than those with no well-perfused parathyroid gland (ICG 2 = 0). Thus, our study demonstrates the feasibility and beneficial role of ICG angiography in transoral endoscopic thyroidectomy.

In our study, 31.7% (19/60) of the inferior parathyroid glands on the lesional side were not seen by visual inspection, which is more as compared with the superior glands (only 11 glands missing). This is because the anatomical location of the inferior parathyroid gland has more variation than the superior one due to difference in embryonic development [18]. In addition, the unique top-down operative view of the transoral approach also made it easier to identify the superior gland once the superior thyroid vessels were ligated and thyroid upper pole was retracted from the deep space. With the assistance of ICG angiography, we were able to identify eight out of the 19 missing inferior parathyroid glands.

Incidental parathyroidectomies occurred in 12 patients, nine of which were due to non-identification, both by the surgeon’s naked eye as well as by ICG angiography. There are three possible explanations for this. First, the feeding vessel of parathyroid gland was injured before the administration of ICG. Thus, the parathyroid gland did not receive any ICG and thus, could not reveal any noticeable fluorescence. Second, when we tried to find a deep-seated parathyroid gland under real-time fluorescence imaging guidance after ICG injection, a vessel injury might have occurred during the soft tissue dissection. Subsequently, the ICG leaked from the injured vessel and spread into the whole operative field, hence it compromised the parathyroid discrimination from surrounding tissue by ICG. Third, the image resolution under the IR mode is relatively poor. Occasionally, the operator had to switch between the white light mode and infrared mode alternatively to evaluate the parathyroid glands.

ICG angiography is non-selective and the fluorescence of a thyroid nodule, fat or lymph node would sometimes be misidentified as a parathyroid gland emission thus leading to the preservation of these tissues (Figure 3) and removal of the actual parathyroid gland. Three parathyroid glands in our study were missed because of this reason. To overcome this problem, autofluorescence has been suggested for the localization of parathyroid gland and high detection rates of 81–100% have been reported [19]. The autofluorescence is generated by the parathyroid gland itself with no need to add contrast agents or targeted labels for fluorescent enhancement, so it is also called label-free identification. However, false positivity of autofluorescence in non-parathyroid tissues, such as brown fat still occurred [20]. In addition, the detection of autofluorescence did not correlate with parathyroid viability because the autofluorescence persisted even after the parathyroid gland was devascularized [21,22]. Moreover, no commercialized laparoscopic equipment is currently available for autofluorescence detection.

Incidental parathyroidectomy is not uncommon in thyroid surgery, with an average rate of 22.4% (range, 16.9% to 43.6%) in a retrospective study consisting of 1114 thyroidectomy and 396 central neck dissection performed by seven surgeons [23]. Accidental parathyroid resection was associated with low surgeon volume, intrathyroidal location of the parathyroid gland, and central neck dissection [23,24]. Two-thirds of the incidental parathyroidectomies that occurred in our study were associated with thyroid malignancies and central neck dissections. Our strategy is to remove all suspicious nodules in the central compartment and incur the risk for unintentional parathyroidectomy, in turn ensuring oncologic safety. Sakorafas et al. stated that incidental parathyroidectomy might not correlate with postoperative hypocalcemia [24]. This further supports the benefit-risk ratio of our strategy.

In one comparative study, the PTH levels on postoperative day were normal in all patients with at least one well-vascularized parathyroid gland [25]. Based on this good predictive power of ICG, a further randomized control trial conducted by the same study group suggested that routine postoperative calcium/PTH measurements and calcium supplementations could be omitted in this group of patients [17]. On the other hand, Rubin et al. found that it took at least two, rather than a single well-perfused gland to better predict the postoperative parathyroid function [16]. Our data revealed that immediate hypoparathyroidism on postoperative day one still occurred in three patients with at least one well-perfused parathyroid gland, but their PTH levels all normalized in one week. Significantly shorter duration of calcium supplementation in patients with at least one well-perfused parathyroid also indicated faster recovery of parathyroid function. In contrast, patients without any well-perfused parathyroid gland all exhibited hypocalcemia on postoperative day one. Thus, routine calcium supplementation might be warranted without delay in this group of patients.

Our studies had two limitations. First, there was no pathological confirmation of the parathyroid glands we assessed. Thus, parathyroid misidentification may be more common than we report. Tissue fluid aspiration from the parathyroid gland for PTH analysis or direct parathyroid gland sampling for frozen section are both feasible ways to validate the identification [10,26]. Nonetheless, either method would inevitably injure the parathyroid glands, and hence was not considered in the present study. Second, despite the fact that we applied the commonly used three-grade scoring system to classify the fluorescence intensity of ICG, a certain degree of arbitrariness is unavoidable. Some authors have used software to measure the fluorescence intensity more objectively; however, this was similar to post-processing of the captured image, rather than real-time quantification [27]. Further improvement of the fluorescence imaging system is expected to overcome this limitation.

## 5. Conclusions

Our study shows that ICG angiography could be a helpful adjunct procedure for parathyroid gland identification and functional outcome prediction during transoral endoscopic thyroidectomy.

## Figures and Tables

**Figure 1 jpm-11-00843-f001:**
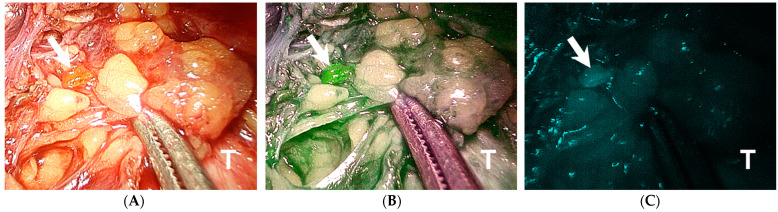
A small left inferior parathyroid gland (arrow) is overlooked during the initial visual inspection (**A**). After indocyanine green injection, the gland is identified by the distinctive fluorescence that contrasted itself from the surrounding soft tissue (**B**,**C**). T, trachea.

**Figure 2 jpm-11-00843-f002:**
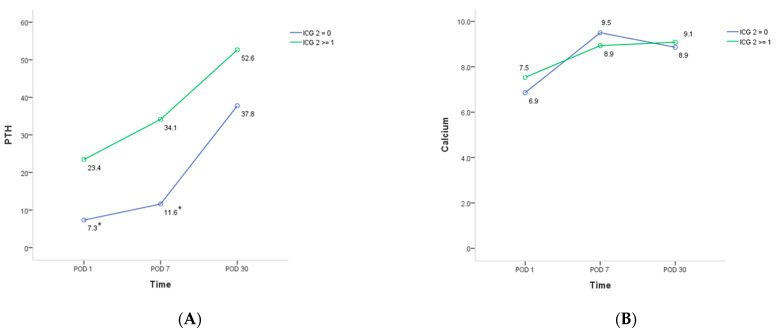
Mean postoperative parathyroid hormone (**A**) and calcium (**B**) levels in patients who underwent total thyroidectomy with or without at least one well-perfused parathyroid gland. Patients who had at least one well-perfused parathyroid gland (ICG 2 ≥ 1) show significantly higher parathyroid hormone level on postoperative day one (POD 1) and after one week (POD 7) than those who had no well-perfused parathyroid gland (ICG 2 = 0) (**A**). The calcium level on postoperative day one is also higher in patients with at least one well-perfused parathyroid gland (ICG 2 ≥ 1), but without statistical significance (**B**). ICG 2, Indocyanine green score 2; POD, Postoperative day; PTH, parathyroid hormone. * statistically significant difference when compared between the two groups (*p* < 0.05).

**Figure 3 jpm-11-00843-f003:**
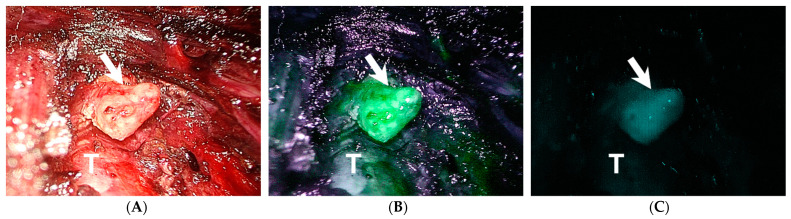
A right paratracheal lymph node (arrow) in a female patient with papillary carcinoma (**A**). It shows significant fluorescent intensity under indocyanine green angiography (**B**,**C**). Fluorescence from other tissues, such as lymph node or fat might sometimes interfere with the detection of parathyroid glands and misidentification can occur. Permanent section of this lymph node revealed reactive lymphadenopathy. T, trachea.

**Table 1 jpm-11-00843-t001:** Demographic and clinical characteristics of patients.

	Patients (*n* = 60)
Age (years)	45.4 ± 12.8
Sex ratio (F:M)	50:10
Body mass index (kg/m^2^)	22.9 ± 3.0
Tumor size (cm)	2.3 ± 1.0
Surgical Procedure	
Lobectomy	41 (68.3%)
Total thyroidectomy	19 (31.7%)
Central neck dissection ^1^	22 (36.7%)
Lymph nodes	
Retrieved	3.0 ± 2.7
Involved	0.9 ± 1.6
Pathology	
Benign	31 (51.7%)
malignant	29 (48.3%)
Hospital stay (day)	3.9 ± 0.5

Data are presented as mean ± SD or number (%). ^1^ Central neck dissection was performed in patients with clinical suspicion for thyroid carcinoma.

**Table 2 jpm-11-00843-t002:** Intraoperative parathyroid scoring with visual inspection followed by indocyanine green angiography.

	Lesion Side ^1^		Contralateral Side	
	Superior Parathyroid (*n* = 60)	Inferior Parathyroid (*n* = 60)	Super Parathyroid (*n* = 19)	Inferior Parathyroid (*n* = 19)
Visual score				
2	48 (80.0%)	41 (68.3%)	18 (94.7%)	18 (94.7%)
0	1(1.7%)	0 (0%)	0 (0%)	0 (0%)
Not identified, initially	11 (18.3%)	19 (31.7%)	1 (5.3%)	1 (5.3%)
Identified after ICG ^2^	1 (1.7%)	8 (13.3%)	0 (0%)	0 (0%)
ICG score				
2	25 (41.7%)	27 (45.0%)	5 (26.3%)	8 (42.1%)
1	22 (36.7%)	21 (35.0%) ^3^	12 (63.2%)	10 (52.6%)
0	3 (5%)	1 (1.7%)	1 (5.3%)	0 (0%)
Not identified	10 (16.7%)	11 (18.3%)	1 (5.3%)	1 (5.3%)

Visual score: V2, well vascularized; V0, no vascularity. Indocyanine green (ICG) score: ICG 2, white, well vascularized; ICG 1, grey, partial vascularized; ICG 0, black, not vascularized. ^1^ For patients who underwent total thyroidectomy, the laterality of the main tumor was the lesion side. ^2^ Parathyroid glands which were not seen by visual inspection at first but were later identified through ICG angiography. ^3^ In three patients, the inferior parathyroid glands detected by visual inspection and ICG angiography were probably misidentified because the parathyroid tissues were found in the resected specimens of thyroid and lymph node.

**Table 3 jpm-11-00843-t003:** Incidentally removed parathyroid glands: incidence, location, cause, and associated thyroid condition.

Parameter	Value
Incidental parathyroidectomy, n	12 (7.6%)
Location	
Superior gland	2 (16.7%)
Inferior gland	10 (83.3%)
Cause of parathyroid gland removal	
Not identified ^1^	9 (75.0%)
Misidentified ^2^	3 (25.0%)
Thyroid pathology	
Benign	4 (33.3%)
Malignant	8 (66.7%)
Surgical procedure	
Lobectomy	8 (66.7%)
Total thyroidectomy	4 (33.3%)
Central neck dissection	8 (66.7%)

^1^ Parathyroid glands that were not identified by both visual inspection and indocyanine green angiography. ^2^ Other tissues, such as fat or lymph node were probably misidentified as parathyroid glands and preserved. The real parathyroid glands were, however, inadvertently removed.

**Table 4 jpm-11-00843-t004:** Comparison between total thyroidectomy patients with at least one well-perfused parathyroid gland (ICG 2 ≥ 1) versus no well-perfused parathyroid gland (ICG 2 = 0).

	ICG 2 = 0 (*n* = 5)	ICG 2 ≥ 1 ^1^ (*n* = 14)	*p*
PTH < 15 (pg/mL) ^2^			
POD 1	4 (80%)	3 (21.4%)	0.038
POD 7	4 (80%)	0 (0%)	0.001
POD 30	0 (0%)	0 (0%)	N.A.
Calcium < 8.4 (mg/dL) ^2^			
POD 1	5 (100%)	11 (78.6%)	0.376
POD 7	0 (0%)	1 (7.1%)	0.737
POD 30	0 (0%)	0 (0%)	N.A.
Incidental parathyroidectomy, *n*	1 (20%)	3 (21.4%)	0.728
Duration of calcium supplement, day	18.2 ± 9.4	8.0 ± 6.1	0.034

^1^ Fourteen patients with at least one well-perfused parathyroid glands: one gland (*n* = 6), two glands (*n* = 5), and three glands (*n* = 3), respectively. ^2^ Normal range at the authors’ institution: PTH (15–68 pg/mL), calcium (8.4–10.6 mg/dL). ICG 2, indocyanine green score 2; N.A., not applicable; POD, postoperative day; PTH, parathyroid hormone.

## Data Availability

The data presented in this study are available on request from the corresponding author.

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
