# Peer review of "Indocyanine Green Angiography for Parathyroid Gland Evaluation during Transoral Endoscopic Thyroidectomy"

_jpm, 2021, doi:10.3390/jpm11090843_

Round 1

Reviewer 1 Report

I read with pleasure the study by TJ Liang and the use of ICG "angiography" for the detection and preservation of parathyroid glands during TOETVA.

The study is very interesting and the data is well analyzed. The results of post-op PTH have to be analyzed with cautious because the study "only" involves 19 patients with total thyroidectomy however it is by far the largest study on ICG and TOETVA in my knowledge and the results are very well described.

I have no comment, very good study indeed, congratulations.

Author Response

Point 1:I read with pleasure the study by TJ Liang and the use of ICG "angiography" for the detection and preservation of parathyroid glands during TOETVA.

The study is very interesting and the data is well analyzed. The results of post-op PTH have to be analyzed with cautious because the study "only" involves 19 patients with total thyroidectomy however it is by far the largest study on ICG and TOETVA in my knowledge and the results are very well described.

I have no comment, very good study indeed, congratulations.

Reponse 1:Thank you for your favorable comment.

We agree that a further study with a larger patient population is warranted to draw a more robust conclusion regarding the effect of ICG angiography on TOETVA.

Reviewer 2 Report

A sample size calculation or a post-hoc power calculation would be useful

Some language editing is needed p.ex. Ln 48: Currently, few study are available... or Ln 90: Then, single dose 90 of... or Fig 2 legend:...but without statistically significant.

Author Response

Point 1:A sample size calculation or a post-hoc power calculation would be useful

Response 1:Thank you for your comment.

This is a preliminary study to assess the effect of ICG angiography in transoral endoscopic thyroidectomy. The data, of a limited number of patients, were collected retrospectively. On the basis of the favorable outcome shown in this study, we plan to conduct a prospective study in the future. The appropriate sample size will be calculated in an attempt to obtain more precise and accurate results regarding the effects of ICG angiography.

Point 2:Some language editing is needed p.ex. Ln 48: Currently, few study are available... or Ln 90: Then, single dose 90 of... or Fig 2 legend:...but without statistically significant.

Reponse 2:Thank you for your comment.

We have used English editing services for language improvement. The changes are indicated in dark red font in the revised manuscript. The corrections made include the following examples:

Currently, few studies are available that have evaluated the implementation of ICG angiography in endoscopic or robotic thyroidectomy [10-12].

(lines 48, page 2)

Then, a single dose of 7.5 mg ICG was administered through peripheral vein and ICG fluorescence was evaluated approximately one minute later by a 30 degree 10 mm infrared (IR) telescope with VISERA ELITE II system (Olympus, Tokyo, Japan) [11].

(lines 90-93, page 2)

The calcium level on postoperative day one is also higher in patients with at least one well-perfused parathyroid gland (ICG 2 ≥ 1), but without statistical significance (B).

(lines 183-184, page 6)